# Effect of Eight-Week Strength Training on Body Composition, Muscle Strength and Perceived Stress in Community-Dwelling Older Women

**DOI:** 10.3390/geriatrics10060136

**Published:** 2025-10-23

**Authors:** Laura Žlibinaitė, Laura Amšiejūtė, Daiva Baltaduonienė, Milda Gintilienė, Karolina Matukynienė, Ligita Mažeikė

**Affiliations:** 1Department of Rehabilitation, Faculty of Medicine, Kauno Kolegija Higher Education Institution, LT-50468 Kaunas, Lithuania; 2Alytus Faculty, Kauno Kolegija Higher Education Institution, LT-62235 Alytus, Lithuania

**Keywords:** muscle mass, elderly, strength exercise, perceived stress, prevention

## Abstract

**Background**: Sarcopenia, characterized by loss of muscle mass, strength, and function, reduces independence and quality of life in older adults. Strength exercise (STR) mitigates these age-related declines, but evidence of short-term effectiveness remains limited. This study aimed to evaluate the effects of eight weeks of STR on body composition, muscle strength, and psycho-emotional state in community-dwelling elderly women. **Methods**: A prospective, controlled, non-randomized study included 44 women assigned to an STR (*n* = 20) or control (CON, *n* = 22) group. The STR group performed supervised exercise twice weekly for eight weeks. The outcomes were body composition, handgrip strength (HGS), quadriceps (Q) and hamstrings (H) strength, and perceived stress (PSS-10), assessed at baseline and after intervention. Within-group changes were analyzed using paired *t*-tests, and between-group differences were evaluated using analysis of covariance (ANCOVA) adjusted for baseline values. **Results**: After baseline adjustment, body mass (*p* = 0.041, partial η^2^ = 0.103), BMI (body mass index, *p* = 0.030, partial η^2^ = 0.115), and body fat percentage (*p* = 0.047, partial η^2^ = 0.098) were significantly reduced in the STR group. Significant improvements were observed for H strength in both legs (*p* < 0.05, partial η^2^ = 0.128–0.131), right HGS (*p* = 0.025, partial η^2^ = 0.122), right HGS:BMI ratio (*p* = 0.013, partial η^2^ = 0.150), and H:Q ratios on both sides (*p* < 0.05, partial η^2^ = 0.109–0.118). No significant differences were observed for left-hand grip strength, knee extensor strength, or other body composition variables (*p* > 0.05). The perceived stress scores were significantly lower in the STR group post-intervention (*p* = 0.036, partial η^2^ = 0.108). **Conclusions**: An eight-week supervised strength exercise program was associated with favorable changes in muscle strength, body composition, and psycho-emotional state in elderly women.

## 1. Introduction

Sarcopenia can be described as an epidemic of an aging society. It is a systemic progressive skeletal muscle disorder characterized by a decrease in muscle mass and strength in elderly population [1]. Sarcopenia’s prevalence varies due to different definitions and diagnostic criteria used. In the literature it is indicated that the prevalence can range from 4.8% to 68.9% [2,3]. Decreased muscle mass, strength, and physical activity have been associated with lower quality of life, together with increased fear of falling and loss of independence and even death [1,4]. However, there is still a lack of sufficient evidence to determine whether strength training is equally effective for older individuals with sarcopenia.

Sarcopenia is caused by many complex and interdependent pathophysiological mechanisms, including aging, physical inactivity, neuromuscular diseases, anabolic insufficiency and insulin resistance, lipotoxicity, endocrine factors [5], oxidative stress, mitochondrial dysfunction, and inflammation [6]. The relationship between psychological stress and the development of sarcopenia in elderly people is complex and versatile. Stress can accelerate the loss of muscle mass and functional activities, leading to the onset or progression of sarcopenia. This relationship is influenced by hormonal changes, nutritional status, and physical activity levels. Therefore, elderly individuals experiencing high stress levels are at greater risk for loss of physical capacity, including falls and fractures [7].

Various exercise protocols have been used to prevent sarcopenia. Resistance training (RT) plays a crucial role in mitigating age-related loss of muscle mass and strength. A 12-week study of progressive RT three times a week showed that it can help slow or even reverse the loss of muscle mass and strength caused by sarcopenia [8]. The effects of RT with resistance bands and free weights on knee extensor and flexor strength were evaluated using an eight-week training protocol performed three times per week. Training intensity was determined using the 10 repetition maximum (RM) method. The results showed that both training protocols were equally effective in increasing training load, as confirmed by the results of the 10 RM tests for knee extension and flexion movements [9]. A resistance exercise program consisting of two exercise sessions per week and involving a combination of upper and lower-body exercises performed with a relatively high degree of effort for 1–3 sets of 6–12 repetitions is appropriate as a treatment for sarcopenia [10].

Body composition is directly related to skeletal muscle mass and strength. Therefore, it is a crucial indicator in evaluating the effect of sarcopenia on daily activities. Bioelectrical impedance analysis (BIA) is used in clinical practice to evaluate body composition quickly and simply [11,12], particularly in cases of various chronic diseases, thereby aiding in the management of sarcopenia. BIA can assess the effectiveness of nutritional and exercise interventions aimed at improving muscle mass and overall health [11].

With advancing age, muscle strength and function decline more rapidly than muscle mass. Before the age of 80, muscle mass decreased more, while muscle strength and function began to decline rapidly over 80 years of age [13]. The strength of the relationship between handgrip strength and muscle moment strength ranges from weak to strong, depending on the assessment methods used and the characteristics of the study population [14].

There is a strong negative significant relationship between leg muscle strength and functional mobility ability in the elderly [15]. Researchers have found that a high body mass index had a protective effect on the reduction in muscle mass in both men and women. However, obesity-related parameters, including body mass index, waist circumference, and body fat percentage, were positively correlated with a lower incidence of sarcopenia only in women. Obesity in older women may have a protective effect in reducing appendicular skeletal muscle mass index and the incidence of sarcopenia [16]. Moreover, intrinsic foot muscle strengthening interventions contributed to improvements in muscular strength, balance, and functional mobility in adults aged 65 years and older, and may have potentially reduced the risk of falls [17,18].

The primary aim of this study was to evaluate the effects of eight weeks of strength training on body composition, handgrip strength, lower limb muscle strength, and psycho-emotional state in community-dwelling elderly women. We hypothesized that strength training will result in significant improvements in muscle mass, handgrip strength, and knee flexion/extension muscle strength, contributing to the overall mitigation of sarcopenia as well as reducing the perceived stress in this population.

This study aims to provide valuable insights into the effectiveness of strength training as a rehabilitation intervention for sarcopenia in older women, a growing public health concern. By demonstrating the potential benefits of strength training, the findings could inform interventions aimed at preserving muscle function and enhancing the quality of life in the elderly population.

## 2. Materials and Methods

### 2.1. Study Design

This study employed a prospective, controlled, non-randomized study design to evaluate the effects of an eight-week strength-focused exercise program on physical fitness, body composition, and psycho-emotional well-being in older adult women. Participants were divided into two groups: an intervention group, which participated in the structured strength exercise program, and a control group, which did not engage in any structured physical activity during the study period.

At baseline, all participants underwent a comprehensive evaluation that included assessments of physical condition and anthropometric parameters. Body composition was measured using BIA. Lower limb strength was assessed through isometric torque testing of knee extension and flexion. Thigh circumference was recorded to monitor changes in upper leg muscle mass, and handgrip strength was measured using an isometric dynamometer as an indicator of upper limb strength. In addition, psycho-emotional well-being was evaluated using the Perceived Stress Scale (PSS-10).

The intervention group participated in supervised exercise sessions twice weekly for eight weeks, with each session lasting 60 min. Sessions were led by a licensed experienced physiotherapist and included a variety of resistance training exercises utilizing dumbbells, resistance bands, and stability balls to target major muscle groups.

Following the eight-week intervention period, post-intervention assessments were conducted for both groups using the same protocols and measurement tools applied at baseline. This design allowed for the assessment of intervention effects by comparing post-intervention outcomes between groups while controlling for baseline values.

### 2.2. Ethical Considerations

The local Biomedical Ethics Committee approved all procedures (Protocol No. 1/24/25). All subjects were given a detailed written explanation of the experimental protocol before they were tested. The participants were informed about the procedure, where the study would be conducted, the study aim, protocol, methods, possible discomforts and risks. They could refuse to participate in the study at any time. Also, they were informed that they participated voluntarily, and all expenses related to participation in this study were not refundable. Furthermore, they were informed how their data would be coded, kept, and used.

### 2.3. Participants

Older community-dwelling women were recruited from Kaunas and Alytus districts. Participants were recruited through Third Age University community, social media, flyers, notices, and word of mouth. Eligibility of interested individuals was assessed during the recruitment process based on the following inclusion criteria: (i) women aged between 65 and 80 years, (ii) have not engaged in ≥150 min of moderate-to-vigorous physical activity per week at baseline, and (iii) capable to move independently. Exclusion criteria included the following: (a) severe exacerbations of chronic diseases, (b) oncological conditions, (c) severe cardiovascular diseases, (d) orthopedic injury or surgery in the last six months, (e) neurological diseases or severe psychiatric illness.

Before initiating the study, a statistical power analysis (software package G*Power 3.0) indicated that a total number of 40 participants (20 participants in each group) were needed for a one-tailed test at 5% significance level with statistical power level of >80%. The numbers were increased to 25 participants per group to account for potential dropouts.

After the initial evaluation, 61 participants were assessed for eligibility. However, 11 participants were excluded for reasons such as not meeting the inclusion criteria, declining to participate, or other unspecified reasons. In total, 50 older women were enrolled in the study. They were assigned either to the Strength exercise group (STR, *n* = 25) or the Control group (CON, *n* = 25). During the intervention period, 5 participants from the Strength exercise group and 3 participants from the Control group dropped out. The final analysis included 20 and 22 participants, respectively. The participants’ characteristics are summarized in Table 1. The flow of participants is presented in the STROBE flow chart (Figure 1). No randomization procedure was applied in this study.

### 2.4. Outcome Measures

#### 2.4.1. Primary

Handgrip strength was chosen as a clinically relevant and sensitive indicator of sarcopenia risk [19]. Upper limb strength was evaluated using a hydraulic hand dynamometer (Jamar, Patterson Medical, Warrenville, IL, USA), a widely used and reliable tool in clinical and research settings. Participants were seated upright with their back supported, feet flat on the floor, and elbow flexed at 90° without arm support. The dynamometer was held in the hand, and participants were instructed to squeeze as hard as possible for 3–5 s. The test was repeated three times for each hand, with brief rest intervals between trials. The mean value from three attempts was used for analysis.

#### 2.4.2. Secondary

Participant height was measured using a portable stadiometer (accuracy ± 0.1 cm).

Body mass and composition were assessed using a segmental bioelectrical impedance analyzer (Tanita MC-780, Tanita Corp., Tokyo, Japan), which estimates body composition by measuring tissue resistance to a low-level electrical current across different body segments (arms, legs, trunk). Prior to the assessment, participants were instructed not to eat or drink for 4 h, not to intensively exercise for at least 12 h, not to consume alcohol for at least 48 h, not to take diuretic medicine for seven days, and to empty their bladder and bowels 30 min before test. During the procedure, participants stood barefoot on the analyzer platform, wearing light clothing and no metal accessories. Height (cm), sex, and age (years) were entered manually, while body weight (kg), BMI (kg/m^2^), body fat (%), muscle mass (kg), bone mineral content (kg), basal metabolic rate (BMR) (kcal/day), and total body water (kg) were recorded automatically.

Thigh circumference was measured using a flexible measuring tape while participants stood upright and relaxed. The measurement was taken at the midpoint between the inguinal crease and the proximal border of the patella, with the tape placed horizontally around the thigh. Care was taken to ensure the tape was snug but not compressing the skin. Three measurements were taken, and the mean value was used for analysis.

Isometric strength of the knee extensors was measured using the handheld dynamometer (K-Push V3, Kinvent Biomechanics, Montpellier, France). Participants were seated on an adjustable bench with their knees flexed at a 90° angle and feet off the ground. The dynamometer was positioned on the anterior aspect of the lower leg, approximately two fingers above the ankle joint. Participants were instructed to push maximally against the device (knee extension) for 5 s while the examiner stabilized the device. Three trials were performed for each leg, with 30 s rest intervals between trials. The highest recorded value was used for analysis. Data was recorded via the device’s associated mobile application.

Isometric strength of the knee flexors was assessed with participants lying prone, the knee flexed to 90°. The dynamometer was placed on the posterior aspect of the lower leg, two fingers above the ankle. The examiner stabilized the pelvis to prevent compensatory movements. Participants were instructed to flex the knee forcefully and rapidly against the dynamometer for 5 s. Three repetitions were performed per leg, with 30 s rest intervals. The best of the three trials was recorded via the connected mobile application.

Psycho-emotional state was measured using the 10-item Perceived Stress Scale (PSS-10), developed by Cohen et al. [20], which assesses the degree to which individuals perceive their lives as unpredictable, uncontrollable, and overloaded. Participants responded to items reflecting their stress-related thoughts and feelings over the past month using a 5-point Likert scale (0 = never to 4 = very often). Positively phrased items were reverse-scored, and a total score was calculated, ranging from 0 to 40, with higher scores indicating higher perceived stress levels.

### 2.5. Intervention

The experimental group participated in an 8-week supervised exercise program, conducted twice per week, with each session lasting 60 min. The program was led by an experienced physiotherapist and aimed to improve muscle strength, stability, and body composition, following American College of Sports Medicine (ACSM, 2019) recommendations for older adults [21]. Each session included exercises using various modalities: bodyweight, resistance bands, free weights, a large stability ball, and a small soft ball. Exercises targeted major muscle groups and incorporated multi-joint movements, focusing on the upper and lower body as well as the core. Sample exercises included bodyweight squats, lunges, planks, bridges on the stability ball, resistance band rows, and dumbbell arm raises. Exercises were performed in three sets of 10–15 repetitions, or held for 30–60 s, with short rest periods between sets. Exercise intensity was monitored using the Borg Rating of Perceived Exertion (RPE) 6–20 scale, with participants instructed to maintain an effort corresponding to 11–14 (“light” to “somewhat hard”). Progression was implemented when participants could exceed 15 repetitions at RPE < 12, by increasing resistance (switching to a higher-tension band color or shortening the band), adding load, or increasing exercise complexity. All sessions were supervised to ensure correct execution and minimize injury risk. A detailed week-by-week sample protocol, including exercise names, sets/reps, rest, and progression rules, is provided in Appendix A to facilitate reproducibility.

Participants in the intervention group attended on average 15 out of 16 scheduled sessions (94% adherence). No adverse events or exercise-related injuries were reported during the 8-week program.

### 2.6. Statistical Analysis

The data were analyzed using IBM SPSS Statistics software (Version 26.0, IBM Corp., Armonk, NY, USA). All data were found to be normally distributed using the Shapiro–Wilk test. Analyses were performed using complete case data; participants with missing baseline or post-intervention values for a given outcome were excluded from the analysis of that outcome. No imputation was performed. Descriptive statistics are reported as mean ± standard deviation (SD). Baseline differences between the control and intervention groups were evaluated using Independent Sample *t*-test.

Between-group comparisons of post-intervention outcomes were conducted using analysis of covariance (ANCOVA), with the post-intervention value as the dependent variable, group (intervention vs. control) as the fixed factor, and the corresponding baseline value as a covariate. Adjusted group means with standard errors (SE) and 95% confidence intervals (CI) were reported. The homogeneity of regression slopes assumption was verified by including the Group × Baseline interaction term in the model, and no significant interactions were found. Homogeneity of variances was assessed using Levene’s test, and normality of residuals was confirmed by visual inspection of Q–Q plots and Shapiro–Wilk tests of studentized residuals.

The level of significance was set at *p* < 0.05. The effect sizes were reported as partial eta-squared (η^2^p) and interpreted as small (0.01), medium (0.06), or large (0.14).

## 3. Results

### 3.1. Handgrip Strength

ANCOVA revealed a statistically significant between-group difference for the right handgrip strength after adjusting for baseline values (*p* = 0.025, partial η^2^ = 0.122), favoring the intervention group. No significant difference was observed for the left handgrip strength (*p* > 0.05) (Table 2).

### 3.2. Body Composition

After adjusting for baseline values, body mass was significantly lower in the intervention group compared with the control group (*p* = 0.041, partial η^2^ = 0.103). Similarly, BMI was significantly reduced in the intervention group (*p* = 0.030, partial η^2^ = 0.115). A significant between-group difference was also observed for body fat percentage (*p* = 0.047, partial η^2^ = 0.098). No statistically significant differences were found for muscle mass, bone mass, basal metabolic rate, total body water, ECW/TBW ratio or thigh circumference (*p* > 0.05). Adjusted means, 95% confidence intervals, and effect sizes are summarized in Table 3.

### 3.3. Muscle Strength

Significant between-group differences were found for knee flexor strength in both legs (*p* < 0.05, partial η^2^ = 0.128–0.131), favoring the intervention group. Knee extensor strength showed no significant differences between groups (*p* > 0.05). After adjusting for baseline values, a statistically significant between-group difference was found for right handgrip-to-BMI ratio (*p* = 0.013, partial η^2^ = 0.150) favoring the intervention group. Significant differences were also observed for hamstring-to-quadriceps ratios on both sides (Left: *p* = 0.028, partial η^2^ = 0.118; Right: *p* = 0.035, partial η^2^ = 0.109), with higher adjusted means in the intervention group. No significant between-group difference was found for left hand handgrip-to-BMI ratio (*p* = 0.724). Adjusted means, 95% confidence intervals, and effect sizes are summarized in Table 4.

### 3.4. Psycho-Emotional State

Participants in the intervention group reported significantly lower PSS-10 scores compared to controls after the program (*p* = 0.036, partial η^2^ = 0.108), indicating a moderate psycho-emotional benefit (Table 5).

## 4. Discussion

This study aimed to evaluate the effects of eight weeks of strength exercise on body composition, muscle strength, and psycho-emotional state in community-dwelling elderly women. Intervention group participants received 16 training sessions within an eight-week period under the supervision of an experienced physiotherapist. The main findings of this study were that strength exercise increased handgrip strength, isometric lower limb muscle strength and improved psycho-emotional state in community-dwelling elderly women. In addition, strength exercise had a positive effect on body composition, including body mass, body fat percentage and body mass index, whereas these parameters compared with the control group.

We observed a side-specific improvement in right-hand grip strength with a medium effect size, favoring the intervention group, whereas the left hand did not differ. This pattern is consistent with well-described dominance effects—dominant hands are typically stronger (often ~10%) and may respond more readily to training in older adults [22]. Exercise programs in elderly adults commonly yield improvements in handgrip strength, which supports the direction of our finding [23]. Given that low muscle strength (assessed via handgrip) is the frontline marker of probable sarcopenia, even unilateral gains may carry clinical relevance [24]. At the same time, emerging work links handgrip asymmetry to functional risk, underscoring the value of reporting laterality and considering handedness in future trials.

The observed improvements in lower limb muscle strength may be explained by several physiological adaptations to regular exercise in older adults, including enhanced neuromuscular activation [25], increased muscle fiber recruitment [26], and improved muscle metabolism [27]. Supervised exercise provides structured and progressive overload, which stimulates muscle hypertrophy and strength gains despite age-related anabolic resistance [28]. The reduction in psycho-emotional stress might be partly attributable to the release of endorphins and other neurochemical mediators of mood during exercise [29], along with the social engagement and sense of achievement experienced during supervised group sessions [30]. The favorable changes in body composition, particularly lower body mass and body fat percentage, may reflect an improved overall energy balance via higher total daily energy expenditure from structured exercise and non-exercise activity thermogenesis (NEAT), rather than an increase in resting metabolic rate (our BMR did not change significantly) [31,32]. This interpretation aligns with recent evidence showing that short-term resistance or combined training often yields modest or no measurable changes in RMR despite beneficial effects on body composition [33]. Minor contributions from hydration-related shifts cannot be excluded, as BIA studies in older adults indicate age- and training-related variation in fluid distribution that can influence composition estimates [34]. Longer randomized trials with objective energy-expenditure and neuromuscular assessments are needed to clarify mechanisms.

The results of this study are consistent with previous research indicating the benefits of structured exercise interventions for older adults. For example, Kong et al. [35] conducted a scoping review of 36 studies and concluded that progressive RT consistently enhances muscle strength in healthy older adults, although effects on muscle morphology are more modest. It is established that structured, progressive resistance exercise programs significantly enhance muscle strength thereby effectively counteracting sarcopenia-associated decline [10]. Similar to our results, Goodarzi et al. [36] confirmed that exercise participation led to significant reductions in stress and enhanced psychological well-being in older populations.

Furthermore, our findings on improved body composition parameters agree with those of Villareal et al. [37], who showed that exercise combined with caloric control improved fat-to-lean mass ratio in obese older adults. Notably, while previous research often emphasizes combining exercise with calorie restriction [38], the present findings indicate that strength exercise alone, even without dietary changes, can yield comparable benefits.

Despite the central role of RT in increasing muscle mass, it has also been shown to reduce fat mass, as evidenced by two separate meta-analyses [39]. In our study, supervised strength exercise was associated with lower body fat percentage, favoring the intervention group, whereas muscle mass did not change over eight weeks period. This pattern is consistent with recent evidence in older adults showing that RT can decrease body fat percentage while increasing limb strength, even in the absence of prescribed dietary interventions [8]. Taken together, these data suggest that short-duration, moderate volume programs may preferentially affect adiposity and neuromuscular function before detectable hypertrophy emerges; longer interventions and/or targeted nutritional advice may be needed to elicit measurable gains in muscle mass.

This study has several strengths, including the supervised nature of the intervention, which ensured adherence, correct technique, and progression of exercises. In addition, the study assessed a comprehensive range of outcomes, encompassing physical, psychological, and anthropometric domains, providing a holistic evaluation of the intervention’s effects. However, some limitations should be noted. The relatively short duration (eight weeks) may not fully capture the long-term effects or sustainability of the observed changes. The sample size was modest, limiting generalizability of the findings. This study used a non-randomized design, with participants self-selecting into the exercise or control groups, which may have introduced selection bias, as more motivated individuals may have chosen to participate in the exercise program. In addition, unmeasured confounders such as nutritional status, daily physical activity outside the program, or psychosocial factors could have influenced the outcomes. Finally, the control group maintained their usual lifestyle rather than participating in an alternative activity, which may have introduced bias due to differences in attention or social interaction.

The findings of this study have potential clinical relevance. They indicate that supervised exercise may be a feasible strategy to promote physical and mental health in older adults living in the community. Implementing similar programs in community centers, nursing homes, or outpatient rehabilitation settings could contribute to healthy aging, lower sarcopenia risk, and enhance quality of life. The simplicity and accessibility of supervised strength exercise suggest that it may be a practical approach for public health initiatives targeting elderly populations.

## 5. Conclusions

Eight weeks of supervised strength exercise were associated with improvements in muscle strength, body composition, and psycho-emotional state among community-dwelling older women, suggesting that such programs may be a feasible short-term strategy to address sarcopenia risk.

## Figures and Tables

**Figure 1 geriatrics-10-00136-f001:**
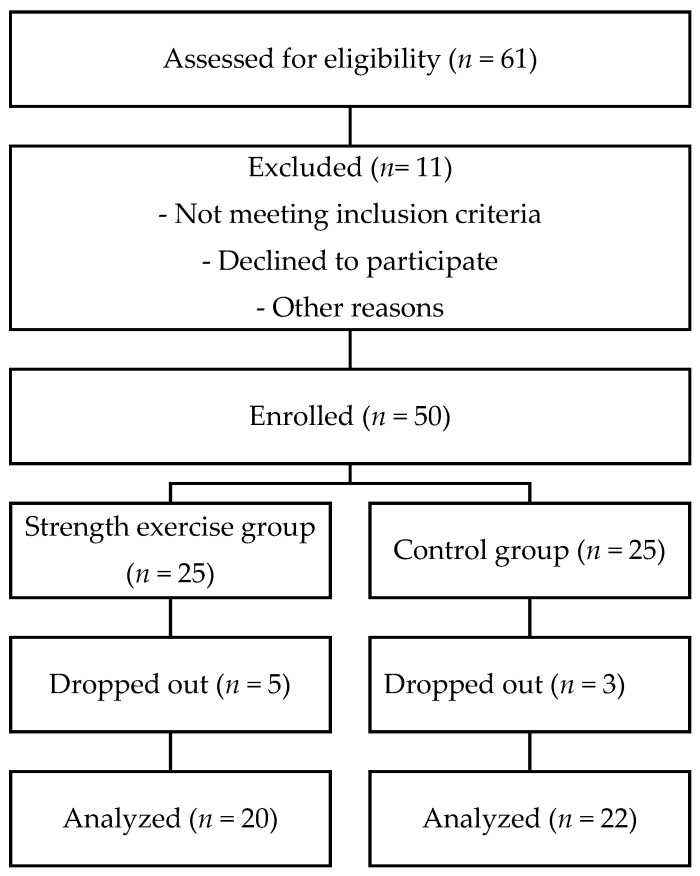
Flowchart of the screening and allocation of the participants.

**Table 1 geriatrics-10-00136-t001:** Baseline characteristics of participants.

Characteristics	STR Group (*n* = 20) Mean ± SD	CON Group (*n* = 22) Mean ± SD	*p*-Value
Age (years)	73.65 ± 6.10	72.14 ± 4.17	0.349
Height (cm)	161.85 ± 5.45	163.73 ± 5.98	0.296
Weight (kg)	67.21 ± 9.40	72.70 ± 11.59	0.101
Body mass index (kg/m^2^)	25.70 ± 3.72	27.09 ± 3.73	0.234
Handgrip strength (kg)			
right	25.98 ± 4.69	27.73 ± 5.21	0.261
left	24.95 ± 4.06	25.70 ± 4.87	0.591

Note: Analyzed by an Independent Samples *t*-test. Abbreviations: STR, strength exercise; CON, control; SD, standard deviation.

**Table 2 geriatrics-10-00136-t002:** ANCOVA-adjusted handgrip strength after the intervention.

Outcome	Adjusted Mean ± SE (STR) [95% CI]	Adjusted Mean ± SE (CON) [95% CI]	STR–CON (95% CI)	F(df1,df2), *p*	Partial η^2^
Handgrip strength (Left), kg	25.34 ± 0.45 [24.43–26.24]	25.33 ± 0.43 [24.47–26.19]	0.01 [−1.24–1.26]	F(1,39) = 0.00, *p* = 0.991	0.000
Handgrip strength (Right), kg	27.70 ± 0.45 [26.79–28.61]	26.25 ± 0.43 [25.38–27.11]	1.46 [0.19–2.72]	F(1,39) = 5.43, *p* = 0.025	0.122

Note: CON, control; CI, confidence interval; SE, standard error; STR, strength exercise.

**Table 3 geriatrics-10-00136-t003:** ANCOVA-adjusted body composition outcomes after the intervention.

Outcome	Adjusted Mean ± SE (STR) [95% CI]	Adjusted Mean ± SE (CON) [95% CI]	STR–CON (95% CI)	F(df1,df2), *p*	Partial η^2^
Body Mass (kg)	69.20 ± 0.30 [68.59–69.81]	70.10 ± 0.29 [69.52–70.68]	−0.90 [−1.76–−0.04]	F(1,39) = 4.46, *p* = 0.041	0.103
BMI (kg/m^2^)	26.10 ± 0.11 [25.87–26.32]	26.44 ± 0.11 [26.23–26.65]	−0.35 [−0.66–−0.03]	F(1,39) = 5.06, *p* = 0.030	0.115
Muscle Mass (kg)	43.10 ± 0.24 [42.62–43.58]	42.55 ± 0.23 [42.08–43.02]	0.55 [−0.13–1.23]	F(1,38) = 2.70, *p* = 0.109	0.066
Body Fat (%)	34.12 ± 0.31 [33.49–34.74]	35.01 ± 0.29 [34.41–35.60]	−0.89 [−1.77–−0.01]	F(1,39) = 4.22, *p* = 0.047	0.098
Bone Mass (kg)	2.32 ± 0.02 [2.29–2.36]	2.29 ± 0.02 [2.26–2.32]	0.03 [−0.02–0.08]	F(1,39) = 1.82, *p* = 0.185	0.045
Basal Metabolic Rate (kcal)	1342.47 ± 7.12 [1328.05–1356.88]	1343.27 ± 6.94 [1329.21–1357.33]	−0.81 [−21.14–19.53]	F(1,38) = 0.01, *p* = 0.937	0.000
Total Body Water (L)	46.04 ± 1.51 [42.99–49.08]	43.91 ± 1.43 [41.01–46.80]	2.13 [−2.14–6.40]	F(1,39) = 1.02, *p* = 0.319	0.025
ECW/TBW Ratio	47.08 ± 0.19 [46.70–47.46]	47.27 ± 0.18 [46.91–47.64]	−0.19 [−0.73–0.34]	F(1,39) = 0.55, *p* = 0.463	0.014
Thigh circumference (Left), cm	50.40 ± 0.37 [49.66–51.15]	50.41 ± 0.35 [49.69–51.12]	−0.00 [−1.04–1.04]	F(1,39) = 0.00, *p* = 0.999	0.000
Thigh circumference (Right), cm	50.34 ± 0.45 [49.43–51.24]	50.19 ± 0.43 [49.33–51.06]	0.14 [−1.12–1.40]	F(1,39) = 0.05, *p* = 0.822	0.001

Note: BMI, body mass index; CON, control; CI, confidence interval; ECW, extra cellular water; SE, standard error; STR, strength exercise; TBW, total body water.

**Table 4 geriatrics-10-00136-t004:** ANCOVA-adjusted muscle strength outcomes after the intervention.

Outcome	Adjusted Mean ± SE (STR) [95% CI]	Adjusted Mean ± SE (CON) [95% CI]	STR–CON (95% CI)	F(df1,df2), *p*	Partial η^2^
Isometric knee flexor strength (Left)	10.87 ± 0.38 [10.10–11.64]	9.61 ± 0.36 [8.88–10.35]	1.26 [0.19–2.32]	F(1,39) = 5.70, *p* = 0.022	0.128
Isometric knee flexor strength (Right)	10.98 ± 0.41 [10.16–11.80]	9.62 ± 0.39 [8.84–10.40]	1.36 [0.23–2.50]	F(1,39) = 5.88, *p* = 0.020	0.131
Isometric knee extensor strength (Left)	22.19 ± 0.81 [20.55–23.82]	22.29 ± 0.77 [20.73–23.84]	−0.10 [−2.47–2.27]	F(1,39) = 0.01, *p* = 0.932	0.000
Isometric knee extensor strength (Right)	23.55 ± 0.73 [22.08–25.02]	22.64 ± 0.69 [21.25–24.04]	0.91 [−1.19–3.01]	F(1,39) = 0.77, *p* = 0.385	0.019
Handgrip-to-BMI Ratio (Left)	0.985 ± 0.019 [0.946–1.024]	0.975 ± 0.019 [0.938–1.013]	0.010 [−0.045–0.064]	F(1,39) = 0.13, *p* = 0.724	0.003
Handgrip-to-BMI Ratio (Right)	1.074 ± 0.018 [1.037–1.111]	1.007 ± 0.018 [0.972–1.043]	0.067 [0.015–0.118]	F(1,39) = 6.86, *p* = 0.013	0.150
Hamstring-to-Quadriceps Ratio (Left)	0.560 ± 0.042 [0.475–0.644]	0.425 ± 0.040 [0.345–0.506]	0.135 [0.015–0.254]	F(1,39) = 5.20, *p* = 0.028	0.118
Hamstring-to-Quadriceps Ratio (Right)	0.494 ± 0.026 [0.442–0.546]	0.417 ± 0.024 [0.367–0.466]	0.077 [0.006–0.149]	F(1,39) = 4.79, *p* = 0.035	0.109

Note: BMI, body mass index; CON, control; CI, confidence interval; SE, standard error; STR, strength exercise.

**Table 5 geriatrics-10-00136-t005:** ANCOVA-adjusted perceived stress scores after the intervention.

Outcome	Adjusted Mean ± SE (STR) [95% CI]	Adjusted Mean ± SE (CON) [95% CI]	STR–CON (95% CI)	F(df1,df2), *p*	Partial η^2^
Perceived Stress Scale (PSS-10)	12.67 ± 0.79 [11.08–14.26]	15.03 ± 0.75 [13.51–16.55]	−2.36 [−4.57–−0.16]	F(1,39) = 4.70, *p* = 0.036	0.108

Note: CON, control; CI, confidence interval; SE, standard error; STR, strength exercise.

## Data Availability

The data presented in this study are available upon request from the corresponding author. The data are not publicly available because, due to the sensitive nature of the questions asked in this study, participants were assured raw data would remain confidential and would not be shared.

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
