# Peer review of "Effect of Eight-Week Strength Training on Body Composition, Muscle Strength and Perceived Stress in Community-Dwelling Older Women"

_geriatrics, 2025, doi:10.3390/geriatrics10060136_

Round 1

Reviewer 1 Report

Comments and Suggestions for Authors

Dear Authors,

Your manuscript addresses a clinically and publicly relevant topic in geriatrics: the interplay between strength training, sarcopenia prevention, and psychosocial well-being in community-dwelling older women. The study’s translational emphasis (a supervised, 8-week program) enhances practical applicability. Strengths include: (i) a focus on sarcopenia with attention to psycho-emotional outcomes; (ii) a short, feasible intervention suitable for community settings; and (iii) a broad outcome set (BIA-based body composition, upper/lower-limb strength, and perceived stress), supporting a holistic appraisal of effects. These features position the work to make a useful, practice-oriented contribution, provided several critical issues are addressed.

MAJOR COMMENTS (MUST ADDRESS)

1) Study design labeling: “prospective matched cohort” is undocumented.

Issue: The manuscript claims a prospective matched cohort design but does not specify any matching protocol (variables, tolerance, ratio, or algorithm). Without this information, the “matched” label is unsupported and misleading.

Action required: If matching was performed, add a dedicated paragraph to Methods detailing:

(a) matching variables (e.g., age tolerance, baseline BMI tolerance, functional status),

(b) matching ratio (1:1, 1:2, etc.), and

(c) the procedure/algorithm (e.g., individual matching, propensity-score matching).

If matching was not performed, remove “Matched” from the title, abstract, and text, and accurately describe the design as a prospective controlled cohort or a non-randomized controlled study. Please revise any language that implies comparability deriving from matching.

2) The reporting framework and terminology are inconsistent with the design

Issue: The inclusion of a CONSORT flow diagram and terms such as “allocated” implicitly suggest randomization, which is not applicable in an observational cohort.

Action required:

a) Replace the CONSORT diagram with a STROBE-consistent cohort flow diagram, showing screening, eligibility, inclusion, follow-up, and analysis by group.

b) Replace terms implying randomization (e.g., “allocated”) with non-randomized language (“enrolled in,” “assigned based on participation,” “grouped by participation”) and state explicitly that no randomization occurred.

3) Primary analysis should adjust for baseline values (ANCOVA)

Issue: Between-group post-test t-tests do not account for baseline variability and are suboptimal for pre-post designs.

Action required:

a) Re-analyze primary outcomes using analysis of covariance – ANCOVA (post-intervention value as the dependent variable; group as fixed factor; the corresponding baseline value as a covariate).

b) Report adjusted group means, effect sizes (e.g., partial η² or Hedges’ g), and 95% confidence intervals.

c) Briefly note ANCOVA assumptions (linearity, homogeneity of regression slopes) and how they were checked.

4) Results: narrative–table inconsistency and a likely data error

Issue A – Narrative vs. Table 3: Section 3.2 states that handgrip strength (HGS) and HGS/BMI remained stable in both groups (p>0.05), yet Table 3 shows between-group differences for right HGS (p=0.016) and right HGS/BMI (p=0.011).

Action required: Revise the text to acknowledge and interpret the between-group differences presented in Table 3, aligning the narrative with the tabulated results.

Issue B – Implausible bone-mass value (Table 2): In Table 2, bone mass (kg) for the intervention group post-intervention is listed as 3.17±4.04 versus 2.27±0.19 at baseline – physiologically implausible over 8 weeks, with an SD far out of proportion to the mean – indicating a likely entry/typing error.

Action required: Audit the raw dataset and correct the bone-mass value; then re-check the full dataset for similar inconsistencies. Document the correction(s) in the Results or a brief data-quality note.

5) Limitations: explicitly address non-randomization and causal language

Issue: The current Limitations section does not sufficiently address the primary limitation: the non-randomized design.

Action required: Expand the Limitations to discuss selection bias (e.g., more motivated participants choosing exercise) and unmeasured confounding, clarifying that causal inference is constrained. Temper any causal phrasing in the Abstract, Results, and Conclusions accordingly.

MINOR COMMENTS (TO STRENGTHEN CLARITY, TRANSPARENCY, AND REPRODUCIBILITY)

A) Intervention reproducibility: Provide a supplementary table with a sample week-by-week protocol: exercise names, sets/reps, rest, and progression rules; specify how intensity was monitored (e.g., Borg RPE) and how elastic-band resistance was progressed.

B) Terminology and abbreviations: In the Abstract and first mentions, write out terms that may be ambiguous to non-specialists (e.g., hamstrings vs. H, handgrip strength vs. HGS). Ensure all abbreviations are defined at first use (e.g., PSS-10, BIA).

C) Adherence and safety: Report session attendance/adherence (e.g., mean % of sessions completed) and any adverse events or explicitly state that none occurred.

D) Missing data handling: Briefly describe how missing data (if any) were handled (complete case, imputation approach, sensitivity checks).

E) Tables/figures and journal style: Ensure table/figure titles, notes, and order of back matter strictly follow Geriatrics author guidelines (e.g., order of Supplementary Materials, Author Contributions, Funding, etc.).

In summary, your topic and translational angle are valuable, and the short supervised program is of practical interest. However, the manuscript requires substantial revisions in design, reporting, analysis, data integrity, and interpretation before it can be considered further. Please address each point above, indicating these revisions, particularly: (I) accurate design labeling/reporting, (II) ANCOVA reanalysis with adjusted estimates and CIs, (III) correction of the bone-mass values and dataset audit, and (IV) alignment of narrative with tables.

Author Response

Thank you very much for taking the time to review this manuscript. Please find the detailed responses below and the corresponding revisions/corrections highlighted in the re-submitted files.

Comments 1: Study design labeling: “prospective matched cohort” is undocumented.

 Issue: The manuscript claims a prospective matched cohort design but does not specify any matching protocol (variables, tolerance, ratio, or algorithm). Without this information, the “matched” label is unsupported and misleading.

 Action required: If matching was performed, add a dedicated paragraph to Methods detailing:

(a) matching variables (e.g., age tolerance, baseline BMI tolerance, functional status),

(b) matching ratio (1:1, 1:2, etc.), and

(c) the procedure/algorithm (e.g., individual matching, propensity-score matching).

If matching was not performed, remove “Matched” from the title, abstract, and text, and accurately describe the design as a prospective controlled cohort or a non-randomized controlled study. Please revise any language that implies comparability deriving from matching.

Response 1: We thank the reviewer for pointing out this important issue.

We acknowledge that no formal matching procedure (variables, tolerance, ratio, or algorithm) was applied neither participants were randomly allocated during the study. To avoid misleading terminology, we have revised the study design description throughout the manuscript. The term “matched” was removed from the title, abstract, and Methods section. The design is now accurately described as a “a prospective, controlled, non-randomized study” and the Methods section (pp. 3-4) was updated to clarify the grouping process and baseline comparability. To avoid any misleading implication, we have removed terms such as “allocated” and replaced them with neutral language.

Updated title: “Strength Exercise Intervention for Sarcopenia Prevention and Psycho-Emotional State in Community-Dwelling Elderly Women”.

Abstract: Methods: A prospective controlled, non-randomized study included 44 women assigned to the STR (n=20) or control (CON, n=22) group.

Comments 2: The reporting framework and terminology are inconsistent with the design

Issue: The inclusion of a CONSORT flow diagram and terms such as “allocated” implicitly suggest randomization, which is not applicable in an observational cohort.

Action required:

a) Replace the CONSORT diagram with a STROBE-consistent cohort flow diagram, showing screening, eligibility, inclusion, follow-up, and analysis by group.

b) Replace terms implying randomization (e.g., “allocated”) with non-randomized language (“enrolled in,” “assigned based on participation,” “grouped by participation”) and state explicitly that no randomization occurred.

Response 2: 

Thank you for this valuable observation. We agree that the previous reporting style could mislead readers by implying randomization. We have made the following changes:

Replaced the CONSORT flow diagram with a STROBE-compliant cohort flow diagram showing screening, eligibility, inclusion, follow-up, and analysis by group (p. 4).

Revised terminology throughout the manuscript by replacing terms implying randomization (e.g., “allocated”) with neutral language such as “enrolled in” or “grouped”.

Added a clear statement in the Methods section indicating that no randomization procedure was performed (p. 4, paragraph 1).

These changes ensure that the reporting framework and terminology accurately reflect the non-randomized, prospective controlled design of the study.

Comments 3: Primary analysis should adjust for baseline values (ANCOVA)

Issue: Between-group post-test t-tests do not account for baseline variability and are suboptimal for pre-post designs.

Action required:

a) Re-analyze primary outcomes using analysis of covariance – ANCOVA (post-intervention value as the dependent variable; group as fixed factor; the corresponding baseline value as a covariate).

b) Report adjusted group means, effect sizes (e.g., partial η² or Hedges’ g), and 95% confidence intervals.

c) Briefly note ANCOVA assumptions (linearity, homogeneity of regression slopes) and how they were checked.

Response 3: We thank the reviewer for this important observation. In the revised manuscript, we have:

Reanalyzed all primary outcomes using analysis of covariance (ANCOVA), with the post-intervention value as the dependent variable, group (intervention vs. control) as the fixed factor, and the corresponding baseline value as a covariate.

Reported adjusted group means ± standard error, 95% confidence intervals, the adjusted mean difference, F-statistics, p-values, and partial η² effect sizes, interpreted as small (0.01), medium (0.06), or large (0.14).

Added a description of the ANCOVA assumptions to the Statistical Analysis section, noting that linearity and homogeneity of regression slopes were verified by including the Group × Baseline interaction term (non-significant), and residuals were inspected for normality using Shapiro–Wilk tests and Q–Q plots.

These changes can be found in the Methods – 2.6. Statistical Analysis section (p. 6) and in the updated Results tables (Tables 2–5).

Comments 4-A: Results: narrative–table inconsistency and a likely data error

Issue A – Narrative vs. Table 3: Section 3.2 states that handgrip strength (HGS) and HGS/BMI remained stable in both groups (p>0.05), yet Table 3 shows between-group differences for right HGS (p=0.016) and right HGS/BMI (p=0.011).

Action required: Revise the text to acknowledge and interpret the between-group differences presented in Table 3, aligning the narrative with the tabulated results.

Response 4-A: We thank the reviewer for pointing out this important inconsistency. In the revised manuscript, we have:

In the revised manuscript, we have reorganized the Results to present the primary outcome – handgrip strength – first. Accordingly, table numbering has been updated and may differ from the numbering cited in the reviewer’s comments.

Reanalyzed handgrip strength (left and right) and handgrip-to-BMI ratios using ANCOVA, adjusted for baseline values, as recommended in Comment 3.

Updated the Results narrative (Section 3.4) to reflect the significant between-group difference observed for right HGS/BMI ratio (F(1,39)=6.86, p=0.013, partial η²=0.150) and to acknowledge the lack of significant difference for the left side (p=0.724).

Ensured that the text, Table 4, and reported p-values are now fully consistent.

These changes are now clearly reflected in the revised Results section (p. 8) and Table 4.

Comments 4-B: Issue B – Implausible bone-mass value (Table 2): In Table 2, bone mass (kg) for the intervention group post-intervention is listed as 3.17±4.04 versus 2.27±0.19 at baseline – physiologically implausible over 8 weeks, with an SD far out of proportion to the mean – indicating a likely entry/typing error.

Action required: Audit the raw dataset and correct the bone-mass value; then re-check the full dataset for similar inconsistencies. Document the correction(s) in the Results or a brief data-quality note.

Response 4-B: Thank you for this important observation. We audited the raw dataset and identified a transcription error in the post-intervention bone-mass value for the intervention group. The value was corrected prior to re-running all statistical analyses, and descriptive as well as inferential results were updated.

As tables now present ANCOVA adjusted group means ± SE (following the reviewer’s earlier suggestions), the original raw means are no longer shown; however, all adjusted values are based on the corrected dataset. 

Comments 5: Limitations: explicitly address non-randomization and causal language

Issue: The current Limitations section does not sufficiently address the primary limitation: the non-randomized design.

Action required: Expand the Limitations to discuss selection bias (e.g., more motivated participants choosing exercise) and unmeasured confounding, clarifying that causal inference is constrained. Temper any causal phrasing in the Abstract, Results, and Conclusions accordingly.

Response 5: We thank the reviewer for this valuable suggestion. In the revised manuscript, we have made changes accordingly (p. 10, paragraph 2):

This study used a non-randomized design, with participants self-selecting into the exercise or control groups, which may have introduced selection bias, as more motivated individuals may have chosen to participate in the exercise program. In addition, unmeasured confounders such as nutritional status, daily physical activity outside the program, or psychosocial factors could have influenced the outcomes.

We have also carefully revised the Abstract, Results, and Conclusions to avoid causal language.

Minor comments

Comment A): Intervention reproducibility: Provide a supplementary table with a sample week-by-week protocol: exercise names, sets/reps, rest, and progression rules; specify how intensity was monitored (e.g., Borg RPE) and how elastic-band resistance was progressed.

Response A): Thank you for this valuable suggestion. We have prepared Supplementary Table S1, which provides a detailed eight-week progressive strength-exercise protocol including exercise names, sets/reps, rest intervals, and progression rules. Additionally, the Intervention section (pp. 5–6) was revised to describe how exercise intensity was monitored using the Borg RPE scale and how resistance-band tension was progressed (by shortening the band or switching to higher-tension bands).

Comment B): Terminology and abbreviations: In the Abstract and first mentions, write out terms that may be ambiguous to non-specialists (e.g., hamstrings vs. H, handgrip strength vs. HGS). Ensure all abbreviations are defined at first use (e.g., PSS-10, BIA).

Response B): We thank the reviewer for this observation. We carefully re-checked the Abstract and the entire manuscript and confirm that all abbreviations, including HGS, Q, H, PSS-10, and BIA, were already written out at first mention. For the reader's convenience there is also a table of abbreviations used in this manuscript (p. 11), as set by the journal template. 

Comment C): Adherence and safety: Report session attendance/adherence (e.g., mean % of sessions completed) and any adverse events or explicitly state that none occurred.

Response C): We thank the reviewer for this valuable suggestion. In the revised manuscript, we have made changes accordingly (p. 6, paragraph 2): “Participants in the intervention group attended on average 15 out of 16 scheduled sessions (94% adherence). No adverse events or exercise-related injuries were reported during the 8-week program.

Comment D): Missing data handling: Briefly describe how missing data (if any) were handled (complete case, imputation approach, sensitivity checks).

Response D): We thank the reviewer for pointing this out. Missing data were handled using a complete case approach: participants with missing baseline or post-intervention values for a given outcome were excluded from the analysis of that outcome. No imputation procedures were performed, as the amount of missing data was minimal and did not affect group comparability. This information has been added to the 2.6. Statistical Analysis section of the revised manuscript (page 6).

Comment E): Tables/figures and journal style: Ensure table/figure titles, notes, and order of back matter strictly follow Geriatrics author guidelines (e.g., order of Supplementary Materials, Author Contributions, Funding, etc.).

Response E): We thank the reviewer for this reminder. We have carefully revised all tables and figure titles, footnotes, and supplementary material captions to conform to the Geriatrics template. The back-matter sections have been reordered and formatted according to journal requirements.

Reviewer 2 Report

Comments and Suggestions for Authors

This study investigates the effects of strength training on sarcopenia-related indicators and the psycho-emotional state in community-dwelling elderly women, a topic of significant practical importance. The study design is relatively complete, implementing a supervised intervention and measuring multidimensional outcome indicators, thereby providing valuable data to the field. However, the manuscript contains several major issues and requires Major Revision before it can be reconsidered for publication.

1.The title, abstract, and methods section all define this study as a 'prospective matched cohort study'. However, the CONSORT flow chart (Figure 1) on page 5 explicitly uses the term 'Randomized'. The authors are requested to clarify the actual study design and unify all related descriptions throughout the manuscript.

2.The presentation of results in the abstract is inappropriate and likely to confuse readers. The abstract provides a p-value representing the significance of a between-group difference, but the accompanying effect size is actually the within-group effect size for the intervention group. The mismatch between the analysis referenced by the p-value and that of the effect size does not conform to scientific reporting standards and prevents readers from accurately assessing the magnitude of the intervention's effect relative to the control group.

3.There is a direct contradiction within the manuscript regarding the results for Handgrip Strength (HGS). The abstract and Table 3 of the results section both show that the improvement in right-hand grip strength in the intervention group was significantly superior to the control group, with a between-group comparison p-value of 0.016. However, the narrative text in the results section (page 7) explicitly states: "handgrip strength... remained stable in both groups (p>0.05)". The authors need to provide further explanation.

4.For key indicators with significant between-group differences, such as hamstring strength (p=0.017 and p=0.041), the tables fail to report the corresponding between-group effect sizes. This prevents readers from judging the actual magnitude of these significant differences.

5.In Table 1, the unit for participant 'Weight' is incorrectly written as 'cm'.

6.The positioning of the '#' symbol in Figure 2 is problematic; please check it.

7.In the Discussion section, the authors suggest that the observed improvements in body composition may stem from an 'increased resting metabolic rate'. However, according to the data in Table 2, the change in the intervention group's Basal Metabolic Rate (BMR) was not statistically significant (p=0.542), which indicates that this explanation by the authors is not supported by their own study data. The authors need to provide further explanation.

Author Response

Thank you very much for taking the time to review this manuscript. Please find the detailed responses below and the corresponding revisions/corrections highlighted in the re-submitted file.

Comments 1: The title, abstract, and methods section all define this study as a 'prospective matched cohort study'. However, the CONSORT flow chart (Figure 1) on page 5 explicitly uses the term 'Randomized'. The authors are requested to clarify the actual study design and unify all related descriptions throughout the manuscript.

Response 1: We thank the reviewer for this important observation. We have carefully revised the manuscript to accurately reflect the study design as a prospective non-randomized controlled study throughout the title, abstract, methods, and discussion. The term “matched” has been removed, as no formal matching procedure was performed.

In addition, the previous CONSORT flowchart has been replaced with a STROBE-consistent cohort flow diagram that accurately represents participant screening, inclusion, group assignment based on participation, follow-up, and analysis. Terms implying randomization (e.g., “allocated”) have been replaced with neutral language (e.g., “enrolled,” “grouped by participation”), and we explicitly state that no randomization occurred.

Comments 2: The presentation of results in the abstract is inappropriate and likely to confuse readers. The abstract provides a p-value representing the significance of a between-group difference, but the accompanying effect size is actually the within-group effect size for the intervention group. The mismatch between the analysis referenced by the p-value and that of the effect size does not conform to scientific reporting standards and prevents readers from accurately assessing the magnitude of the intervention's effect relative to the control group.

Response 2: Thank you for this important observation. Following the reviewers’ recommendations, we re-analyzed all outcomes using ANCOVA, with post-intervention values as the dependent variable, group as the fixed factor, and baseline values as covariates. We then revised the Abstract Results section to ensure that both the p-values and effect sizes reported are derived from the same analysis. Specifically, we now present between-group adjusted results from ANCOVA together with partial η² as the effect size.

Comments 3: There is a direct contradiction within the manuscript regarding the results for Handgrip Strength (HGS). The abstract and Table 3 of the results section both show that the improvement in right-hand grip strength in the intervention group was significantly superior to the control group, with a between-group comparison p-value of 0.016. However, the narrative text in the results section (page 7) explicitly states: "handgrip strength... remained stable in both groups (p>0.05)". The authors need to provide further explanation.

Response 3: We thank the reviewer for pointing out this inconsistency. The discrepancy arose from an earlier version of the analysis and narrative text. In the revised manuscript, we have re-analyzed handgrip strength using ANCOVA (adjusting for baseline values) and updated the narrative text in Section 3.1 to align with the results presented in the tables and abstract. The text now clearly reports that right handgrip strength showed a statistically significant between-group difference favoring the intervention group, while left handgrip strength did not differ significantly between groups (Table 2). This ensures full consistency between the narrative, tables, and abstract.

Comments 4: For key indicators with significant between-group differences, such as hamstring strength (p=0.017 and p=0.041), the tables fail to report the corresponding between-group effect sizes. This prevents readers from judging the actual magnitude of these significant differences.

Response 4: We appreciate the reviewer’s valuable feedback. In the revised manuscript, we have added partial η² effect sizes for all between-group comparisons to Tables 2–5, including hamstring strength and other outcomes with significant differences. 

Comments 5: In Table 1, the unit for participant 'Weight' is incorrectly written as 'cm'.

Response 5: Thank you for noticing this error. We have corrected the unit for “Weight” in Table 1 to kilograms (kg) in the revised manuscript.

Comments 6: The positioning of the '#' symbol in Figure 2 is problematic; please check it.

Response 6: Thank you for noting this. To improve clarity and ensure consistency within the manuscript, we have replaced the Figure 2 with a table presenting the perceived stress data (now Table 5). 

Comments 7: In the Discussion section, the authors suggest that the observed improvements in body composition may stem from an 'increased resting metabolic rate'. However, according to the data in Table 2, the change in the intervention group's Basal Metabolic Rate (BMR) was not statistically significant (p=0.542), which indicates that this explanation by the authors is not supported by their own study data. The authors need to provide further explanation.

Response 7: We appreciate the reviewer’s comment. We have revised the Discussion section to remove the suggestion that changes in body composition were driven by increased resting metabolic rate. Instead, we now interpret these findings in the context of higher total daily energy expenditure (structured exercise + NEAT) and neuromuscular/muscular adaptations, noting that our BMR did not change significantly. Relevant recent references have been added.

Page 9, paragraph 3:

“<…> The favorable changes in body composition, particularly lower body mass and body fat percentage, may reflect an improved overall energy balance via higher total daily energy expenditure from structured exercise and non-exercise activity thermogenesis (NEAT), rather than an increase in resting metabolic rate (our BMR did not change significantly) [31,32]. This interpretation aligns with recent evidence showing that short-term resistance or combined training often yields modest or no measurable changes in RMR despite beneficial effects on body composition [33]. Minor contributions from hydration-related shifts cannot be excluded, as BIA studies in older adults indicate age- and training-related variation in fluid distribution that can influence composition estimates [34]. Longer randomized trials with objective energy-expenditure and neuromuscular assessments are needed to clarify mechanisms.

Round 2

Reviewer 1 Report

Comments and Suggestions for Authors

Dear Authors,

I would like to congratulate you on the excellent work in revising the manuscript "Strength Exercise Intervention for Sarcopenia Prevention and Psycho-Emotional State in Community-Dwelling Elderly Women." All the concerns raised during the review process have been addressed thoroughly and meticulously.

The changes made, including correcting the study design to a "prospective, controlled, non-randomized study," replacing the CONSORT flowchart with a STROBE diagram, re-analyzing the data with ANCOVA to adjust for baseline variables, and strengthening the discussion of the study's limitations, have significantly improved the quality and rigor of the article. The inclusion of the supplementary table with the detailed exercise protocol has also enhanced the study's reproducibility.

The manuscript is now much more robust, clear, and methodologically sound. Due to the comprehensive nature and high quality of your revisions, I have recommended the article's acceptance for publication.

Author Response

Dear Reviewer,

We sincerely thank you for your positive and encouraging feedback on our revised manuscript. We greatly appreciate your recognition of the changes made and your recommendation for acceptance. Your supportive comments are highly motivating for our team.

Reviewer 2 Report

Comments and Suggestions for Authors

The author has made revisions, but the conclusion still mentions "Two months". I suggest unifying the time frame to "Eight weeks".

Author Response

Dear Reviewer,

Thank you for your careful reading of our manuscript and for pointing out the inconsistency in the time frame. We appreciate your suggestion and have unified the terminology to “eight weeks” throughout the manuscript as advised.